# Ectopic Overexpression of Maize Heat Stress Transcription Factor *ZmHsf05* Confers Drought Tolerance in Transgenic Rice

**DOI:** 10.3390/genes12101568

**Published:** 2021-10-01

**Authors:** Weina Si, Qizhi Liang, Li Chen, Feiyang Song, You Chen, Haiyang Jiang

**Affiliations:** 1National Engineering Laboratory of Crop Stress Resistance Breeding, School of Life Sciences, Anhui Agricultural University, Hefei 230036, China; Weinasi@ahau.edu.cn (W.S.); dg20300024@smail.nju.edu.cn (Q.L.); cl0226@stu.ahau.edu.cn (L.C.); songfy1026@163.com (F.S.); cy19720386@stu.ahau.edu.cn (Y.C.); 2State Key Laboratory of Pharmaceutical Biotechnology, School of Life Sciences, Nanjing University, Nanjing 210093, China

**Keywords:** *ZmHsf05*, maize, drought, rice, abscisic acid

## Abstract

Drought is a key factor affecting plant growth and development. Heat shock transcription factors (*Hsfs*) have been reported to respond to diverse abiotic stresses, including drought stress. In the present study, functional characterization of maize heat shock transcription factor 05 (*ZmHsf05)* gene was conducted. Homologous analysis showed that *ZmHsf05* belongs to Class A2 *Hsfs*. The mRNA expression level of *ZmHsf05* can be affected by drought, high temperature, salt, and abscisic acid (ABA) treatment. Ectopic overexpression of *ZmHsf05* in rice (*Oryza sativa*) could significantly enhance the drought tolerance. Faced with drought stress, transgenic rice exhibited better phenotypic performance, higher survival rate, higher proline content, and lower leaf water loss rate, compared with wild-type plant Zhonghua11. Additionally, we assessed the agronomic traits of seven transgenic rice lines overexpressing *ZmHsf05* and found that *ZmHsf05* altered agronomical traits in the field trials. Moreover, rice overexpressing *ZmHsf05* was more sensitive to ABA and had either a lower germination rate or shorter shoot length under ABA treatment. The transcription level of key genes in the ABA synthesis and drought-related pathway were significantly improved in transgenic rice after drought stress. Collectively, our results showed that ZmHsf05 could improve drought tolerance in rice, likely in an ABA-dependent manner.

## 1. Introduction

The uneven distribution and overall shortage of water resources, as well as increasingly severe global warming, have made drought stress one of the most serious threats to plant survival. Moreover, drought can also exacerbate the effects of other biotic and abiotic stresses, including pests, diseases, extreme temperatures, ion stress, etc [1], seriously affecting growth and yield of important crops, such as maize (*Zea mays*) and rice (*Oryza sativa*) [2]. It has been reported that drought affects grain yields in a quarter of the world’s arable land [3]. Therefore, deciphering the genetic mechanism of plant responses to water deficit and exploring drought-responsive genes would be of great value.

Due to their sessile properties, plants have evolved systemic and effective mechanisms to resist drought stress. Plants respond to drought stress through the integration of sophisticated signaling pathways, resulting in external morphological changes, including changes to plant size, leaf morphology [4,5,6], stomatal characteristics [7,8] and root development [9,10]. This eventually leads to the coordination of plant growth and development. Abscisic acid (ABA) is an important hormone that plays a central role in drought-stress related signaling pathways [11,12]. Upon drought stress, ABA rapidly accumulates, giving rise to leaf stomata closure and the activation of many stress-related genes. Thus far, quite a few key transcription factors have been reported to be involved in response to drought stress through ABA-dependent pathways. For example, an abscisic acid, stress and ripening (*ASR*) gene, *OsASR5* from an upland rice variety can enhance drought tolerance in rice by regulating ABA biosynthesis and promoting stomatal closure [13]. Additionally, *GhWRKY17* negatively regulates drought and salt stress through ABA signaling [14]. In *Arabidopsis thaliana*, the transcription factor *NGATHA1 (NGA1)* could enhance resistance to drought stress through the transcriptional activation of *NINE-CIS-EPOXYCAROTENOID DIOXYGENASE 3 (NCED3)*, which encodes a critical enzyme for ABA biosynthesis [15,16]. These drought-responsive transcription factors are present in a considerable quantity or variety [17]. They generally function through the transcriptional regulation of downstream genes, mainly including osmotic adjustment genes, antioxidant metabolism genes and stress-inducing protein genes [18,19,20,21,22].

Heat shock transcription factors *(Hsfs)* are critical components in signal transduction networks that transcriptionally regulate genes responsive to diverse stresses; they are evolutionarily conserved throughout the eukaryotic kingdom [23]. In vertebrates, Drosophila and *Caenorhabditis elegans*, four, one and one *Hsfs* have been characterized, respectively [24]. In plants, *Hsfs* families were identified to have diverse gene numbers varying among species, with 21 in Arabidopsis, 25 in rice, 24 in tomato (*Solanum lycopersicum*) and 52 in soybean (*Glycine max*), respectively [23,25,26]. Moreover, *Hsfs* in plants shared a typical modular structure [23,27,28]. The N-terminal DNA-binding domain (DBD) features by a central helix-turn-helix that specifically recognizes and binds to the heat stress promoter elements (HSE) (5′-AGAAnnTTCT-3′). The oligomerization domain (OD or HR-A/B region) is linked to the DBD by a flexible linker of variable length. According to the length of the flexible linker and the structural characteristics of the HR-A/B region, Hsfs are assigned to A, B, and C classes. Class A Hsfs generally function as transcriptional activators that are mediated by the C-terminal short activator peptide motifs (AHA motifs), while typical repressor domains are found in class B Hsfs. In addition, the nuclear localization signal (NLS) for nuclear import and the nuclear export signal (NES) for the receptor-mediated nuclear export are also present in Hsfs. 

In line with its nomenclature, the existing functional characterization of *HSFs* is mainly concentrated in the heat stress-related pathway and were well-documented in several model plants, including Arabidopsis and tomato [29,30]. In addition to heat stress, the importance role of *Hsfs* in distinct stress, including drought stress, has begun to receive attention. *AtHsfA6* is transcriptionally regulated by ABA-responsive element binding factor/ABA-responsive element binding protein. Arabidopsis overexpressing *AtHsfA6* shows enhanced salt and drought stress tolerance and is hypersensitive to ABA [31]. The combined overexpression of sunflower *HaHsfA4* and *HaHsfA9* results in a synergistic functional effect on resistance to severe dehydration stress in seedlings [32]. *TaHsfA6f* from wheat could enhance tolerance to drought, and salt stresses of transgenic Arabidopsis [33]. The overexpression of the *OsHSFA3* gene in Arabidopsis can enhance drought tolerance by affecting the levels of ABA and reactive oxygen species [34]. Drought-responsive *HSF* in *Malus domestica*, designated *MdHSFA8a*, was demonstrated to positively modulate flavonoid synthesis in order to regulate drought tolerance [35].

Maize is one of the most important cereal crops across the world. In maize, 31 non-redundant *Hsf* genes have been identified and were assigned to different classes [36]. Recent research about *ZmHsfs* has also mainly focused on heat tolerance. *ZmHsf04* (*ZmHsfA2*) and *ZmHsf01* can improve thermotolerance in transgenic Arabidopsis [37,38]. In the present study, *ZmHsf05* was found to be responsive to drought and could improve the drought tolerance in rice. The present study would provide a gene resource for drought-tolerance breeding in rice and maize.

## 2. Materials and Methods

### 2.1. Gene Structural Features and Protein Sequence Alignments

We obtained the full-length CDS sequence of the *ZmHsf05* gene from the online database Phytozome (https://phytozome.jgi.doe.gov/pz/portal.html accessed on 18 April 2014). To identify all Hsfs from maize, rice and Arabidopsis, proteome of maize were downloaded from MaizeGDB (https://maizegdb.org/, accessed on 1 August 2017), and proteomes of rice and Arabidopsis were downloaded from the Phytozome database. Subsequently, a pfam_scan perl script from HMMER3.1 (Maryland, USA) was used to query all of the three proteomes against the Pfam library to detect the candidate Hsfs harboring HSF_DNA-binding domain (PF00447) [39]. The full-length amino acid sequences of all Hsf proteins in three species were aligned and used to construct a neighbor-joining phylogenetic tree by MEGA 7.0(Auckland, New Zealand) [40]. A bootstrap analysis was performed by 1000 replicates. Multiple alignment of ZmHsf05, OsHsfA2e (Os03g58160) and AtHsfA2 (At2g26150) were conducted in MEGA 7.0 and subsequently submitted to ESPript v3.0 (Lyon, France) [41]. Predicted secondary structure of AtHsfA2 were downloaded from AlphaFold Protein Structures Database [42].

### 2.2. Plant Materials, Growth Conditions, and Transformation

The seeds of maize inbred line B73 used in this study came from the resources preserved in our laboratory. Seeds were soaked and planted in nutrient-rich soil in a plant incubator at 28 ℃ under 14 h light and 22 °Cunder 10 h dark. For tissue specific expression analysis of *ZmHsf05*, leaves, roots, stems, tassels, corn silk, and bract were sampled from B73 seedlings at the adult stage. The rice variety used in the present study was ZH11 (Zhonghua 11), obtained from China Rice Data Center (https://ricedata.cn/variety/varis/601422.htm?601422 accessed on 1 August 2005). Rice was grown in nutrient-rich soil or nutrient solution in plant incubators for experimentation at seedling stages at 26 °C under 14 h light and 22 °C under 10 h dark. The rice nutrient solution we used was Hoagland nutrient solution diluted-5-times. For field experiment, transgenic rice lines, together with wild-type plants, were grown in a paddy field in Anhui agricultural university (Hefei, China).

In the analysis of stress-induced expression patterns, B73 seedling at the three-leaves stage were treated with different abiotic stress, including ABA presence (spraying corn plant leaves with 100 μM of ABA), simulated drought stress (20% polyethylene glycol 6000) and salinity stress (200 mM NaCl), and heat stress (42 °C). Leaves were sampled at 0, 1, 3, 6, 12, and 24 h. For the analysis of stress-related gene expression, rice seedlings were grown in Hoagland nutrient solution for three weeks and then treated with 20% PEG. After 3 h and 6 h of treatment, leaves of ZH11 and transgenic lines were quickly sampled. All sampled tissues were treated by liquid nitrogen, then refrigerate and store at −80 °C for further analysis.

To generate the transgenic rice lines overexpressing *ZmHsf05*, the full-length coding sequence of *ZmHsf05* cloned from B73 was digested with *Kpn*I and *Xba*I, and inserted into the pCAMBIA1301a vector driven by the cauliflower mosaic virus (CaMV) 35S promoter. Moreover, a *β*-glucuronidase (GUS) gene was also under the control of CaMV 35S promoter. Primers are listed in Appendix A. The recombinant plasmid were transferred into *Agrobacterium tumefaciens* strain GV3101 and the Agrobacterium-mediated methods were applied to transform the 35S::*ZmHsf05* recombinant plasmid into rice embryonic calli [43]. To obtain positive transgenic lines, hygromycin screening and GUS straining were performed referring to the previous study [44]. Candidate positive transgenic lines were further confirmed by RT-PCR by detecting the expression of *ZmHsf05* in ZH11 and transgenic lines. Homozygous T3 progeny transgenic lines were utilized for further phenotypic assessment.

### 2.3. Total RNA Extraction and RT-qPCR Analysis

All corn and rice RNA were extracted using the TRIzol reagent (Invitrogen). All template cDNAs were obtained using the TaKaRa’s reverse transcription kit. The RT-qPCR reaction program was: 95 °C 5 min, followed by 40 cycles of 95 °C for 10 s, 60 °C for 30 s. We added a dissolution profile steps after all cycles. The amplified signal and data were processed using the comparative Ct method (ΔΔCt method) [45]. *ZmActin1* (*Zm00001d010159*) and *OsActin1* (*Os03g50885*) were used as controls. Each quantitation experiment included 3 biological replicates and 3 technical replicates. The primer sequences used in this study are shown in Appendix A.

### 2.4. Phenotypic Analysis

For phenotypic analysis of transgenic rice under drought stress, rice seedlings were cultivated in nutrient-rich soil for three weeks in normal condition and withheld water for 7 days. Seedlings were subsequently rewatered for 3 days to observe the phenotype. For the stimulated drought experiments, rice seedlings cultured in the nutrient solution for three weeks and then treated with 20% PEG6000 for 12 h. Treated seedlings were further restored with the nutrient solution for 5 days.

### 2.5. Measurement of Physiological Indices

The water loss rate was estimated according to the methods described by Wei et al. [46]. Leaves of the seedlings in the same position were detached and their fresh weight was measured. Then, the detached leaves were exposed in the air to induce dehydration. Desiccated weight was assessed at 1 h, 2 h and 3 h, respectively. Water loss rate (%) = (fresh weight – desiccated weight)/FW × 100. Proline content was determined following the proline assay kit (A107-1-1, Nanjing Jiancheng Bioengineering Institute, Nanjing, China), with homogenized 0.1 mg of homogenized fresh leaves. Absorbance values were measured at 520 nm.

### 2.6. Subcellular Localization Analysis

We constructed the full-length *ZmHsf05* CDS with the terminator removed on the pCAMBIA1305 vector which has a GFP flag and a CaMV 35S promoter by adding two restriction sites (*Xba I* and *Sma I*). Primers were listed in Appendix A. Maize protoplasts were prepared using the leaves of two-week-old etiolated seedlings according to the method reported by Gao et al. [47]. The resulting ZmHsf05-GFP recombinant plasmid, positive control 35S::GFP, and nucleus marker NLS-mcherry were placed into the prepared maize protoplasts by PEG-calcium mediated method, followed by a 18 h incubation in the dark at 28 °C for transient expression. The fluorescence signals of the recombinant proteins in protoplast cells were observed by a the confocal laser scanning microscope (ZEISS LSM 880 with Airyscan, Germany) with GFP (488 nm excitation, 522–572 nm emission) and mCherry (587 nm excitation, 590–630 nm emission).

### 2.7. Transactivation Activity Assays

The Full-length CDS sequence of *ZmHsf05* was fused to the pGBKT7 vector via homologous recombination. PGBKT7-p53 + pGADT7-T was used as a positive control, and pGBKT7 was used as a negative control. Additionally, 1 × TE/LiAc was used to turn yeast strain AH109 into competent yeast. PEG/LiAc, DMSO, 1 × TE buffer, and salmon sperm DNA were used to transform the constructed plasmid into yeast competent cells. Single-deficient medium SD/-Trp and triple-deficient medium SD/-Trp/-His/-Leu were purchased from TakaRa company.

### 2.8. Statistical Analysis

Fisher’s least significant difference multiple test or Student’s *t*-test was performed via the DPS software to identify the statistical significance of the data [48].

## 3. Results

### 3.1. Molecular Characterization of ZmHsf05 Sequence

A previously identified *Hsf* gene [49] GRMZM2G132971 (B73 v4 gene ID: *Zm00001d034433*), named as *ZmHsf05*, was investigated in the present study. The coding sequence (CDS) of ZmHsf05 is 1080 bp in length and encodes a protein harboring an HSF_DNA-bind domain (PF00447). Moreover, homologous analysis was performed through phylogenic reconstruction with all Hsfs proteins from maize, rice, and Arabidopsis (Figure 1A). Results showed that ZmHsf05 was clustered within the HsfA2 clade and showed an orthologous evolutionary relationship with OsHsfA2e. The domain composition and signature of both the OsHsfA2e (Os03g58160) and AtHsfA2 (At2g26150) has been well characterized. Thus, to better analyze the protein structure of ZmHsf05, protein sequence of ZmHsf05 was aligned to AtHsfA2 and OsHsfA2e (Figure 1B). As expected, ZmHsf05 showed high sequence similarity with AtHsfA2 and OsHsfA2e. Additionally, amino acid sequences and position of N-terminated DNA binding domain (DBD), oligomerization domain (OD), nuclear localization signal (NLS), and aromatics hydrophobic and acidic amino acid residues (AHA) and nuclear export signal (NES) of ZmHsf05 were shown in Figure 1B, according to previous annotation of AtHsfA2 and OsHsfA2e (Figure 1B). Those typical domains were conserved among the three proteins, while only sequences of NES were variable.

### 3.2. Tissue-Specific and Stress-Induced Expression Pattern of ZmHsf05

The tissue-specific and stress-induced expression profiles of *ZmHsf05* were further investigated. We first measured the expression patterns of *ZmHsf05* in various tissues of maize under normal conditions using a quantitative real-time polymerase chain reaction (RT-qPCR), including the root, stem, leaf, tassel, corn silk, and bract. The results showed that *ZmHsf05* was expressed in all surveyed tissues and organs. In stems and leaves, ZmHsf05 showed the highest expression level (Figure 2A). Previous reports have implied that *Hsfs* may be related to abiotic stresses; thus, an expression profiles analysis was further conducted to investigate whether *ZmHsf05* can be induced by various types of abiotic stress. Similarly, we used RT-qPCR to determine the transcription levels of *ZmHsf05* under distinct abiotic environmental stresses, including high temperature, ABA, salt, and drought treatments (Figure 2B–E). Specifically, 20% polyethylene glycol 6000 (PEG 6000) was used to simulate drought stress. The results showed that *ZmHsf05* can be induced in maize by high temperature, ABA, salt, and drought in maize at different time points after treatment. Overall, ZmHsf05 is mainly expressed in the stems and leaves of maize, and can be differentially induced by various abiotic stresses.

### 3.3. Subcellular Location Analysis of ZmHsf05 

The subcellular location of the ZmHsf05 protein was further investigated through transiently expression in the maize protoplast system. The coding sequences of ZmHsf05 were cloned and fused to the N-terminal of green fluorescent protein (GFP) under the control of the CaMV 35S promoter. Then, the recombinant vector was transformed into maize protoplasts. Sixteen hours after transformation, the subcellular location of ZmHsf05 was observed under the confocal laser scanning microscope. As shown in Figure 3, ZmHsf05 was localized in the nucleus, suggesting that ZmHsf05 protein may function in the nucleus.

### 3.4. ZmHsf05 Has Transactivation Activity in Yeast

To investigate whether *ZmHsf05* has transactivation activity, a yeast assay system was used. The full-length CDS of *ZmHsf05* was fused to the GAL4 DBD in the pGBKT7 vector. The recombinant plasmid was then transformed into the AH109 yeast strain (Figure 4A). At the same time, pGBKT7-53+pGADT7-T were used as positive controls and the empty vector pGBKT7 was used as a negative controls. As shown in Figure 4B, the positive controls and pGBKT7-*ZmHsf05* can grow on SD/-Trp single deficiency medium and SD/-Trp/-His/-Leu/-X-α-gal triple deficiency medium. They can also turn blue on the SD/-Trp/-His/-Leu/-X-α-gal triple deficiency medium. The negative control can grow on SD/-Trp single deficiency medium but not on the SD/-Trp/-His/-Leu/-X-α-gal triple deficiency medium. Negative controls cannot turn blue either. Based on the above results, we can observe that *ZmHsf05* exhibits transactivation activity in yeast.

### 3.5. Overexpression of ZmHsf05 in Rice Alters Agronomical Traits in the Field Trials

To better investigate the important roles of *ZmHsf05* in response to environmental stress, *ZmHsf05* was overexpressed in rice cultivar Zhonghua11 (ZH11) driven by the strong and constitutively active 35S promoter. The transgenic rice was generated by Agrobacterium-mediated transformation and confirmed by hygromycin resistance, the positive staining of the GUS marker, and amplifying the sequences of the recombinant vector via PCR. Finally, seven transgenic lines were obtained (Appendix A). In addition, we also detected the expression of the *ZmHsf05* in transgenic rice using quantitative and semi-quantitative methods (Appendix A).

Seven independent transgenic lines and the wild-type plant ZH11 were grown in field condition. Their agronomic traits were thoroughly analyzed to estimate whether overexpressing *ZmHsf05* in rice would influence the growth and development in the field trials. The plant height of six transgenic rice lines was significantly higher than that of the wild-type plants (Figure 5C). Moreover, transgenic rice lines have more effective tillers (Figure 5D), longer panicles (Figure 5A,E) than those of ZH11. Additionally, grains of transgenic lines were approximately equal to those of wild-type plants. Whereas, grains of transgenic lines were a little bit longer, but thinner (Figure 5B,F,G). Intriguingly, the 1000-seed weight were significantly reduced in four transgenic lines compared to wild type plants, the other three transgenic lines also showed similar decreasing tendency (Figure 5H). The results demonstrated that overexpression of *ZmHsf05* in rice can promote effective tiller emergence, improve plant height, increase panicle length, modify seed morphology and cause serious damage to seed weight. The transgenic lines have a considerably larger number of effective tillers, which suggested overexpressing *ZmHsf05* in rice may have great potential to increase yield production.

### 3.6. Overexpression of ZmHsf05 Enhances Drought Tolerance in Rice

Subsequently, in order to investigate whether ZmHsf05 can improve the drought tolerance of transgenic rice, we selected three transgenic rice lines for drought experiments. The three transgenic rice lines with *ZmHsf05* overexpression lines (OE1, OE2, OE3) and ZH11 were cultured in soil. Three weeks after planting, the rice lines had water withheld water for about 7 days, resulting in the leaves wilting. Then, the surveyed rice lines were watered and allowed to recover for 3 days. As shown in our results, the transgenic rice lines showed better phenotypic performances after exposure to drought treatment for 7 days or recovering for 3 days (Figure 6A–C). Moreover, the survival rate of the transgenic rice and ZH11 further verified our phenotypic results (Figure 6D). After the drought treatment, the survival rate of transgenic rice was significantly higher than that of ZH11.

In addition, we used PEG to simulate the drought treatment of transgenic rice. After three weeks of culturing in five-times-diluted Hoagland nutrient solution, we used 20% PEG to simulate drought treatment for 12 h and then restored the culture with five- times-diluted Hoagland nutrient solution for 5 days. Before the simulated drought treatment, the growth status of transgenic plants and wild-type ZH11 were similar; however, after 12 h of simulated drought and cultivation had been resumed for 12 h, the growth status of transgenic rice was apparently evidently better than that of ZH11 (Figure 7A).

Free proline is an important metabolite in drought stress, benefiting the maintenance of cellular homeostasis upon environmental stress. The accumulation and mobilization of proline are believed to be coincidental with the trait of drought tolerance [46]. To further decipher the physiological basis underlying drought tolerance in transgenic rice, the proline contents of ZH11 and transgenic rice were measured before and after the simulated drought treatment. We found that before drought treatment, the proline content in transgenic rice plants was approximately equal to that of ZH11. Whereas, the proline content in transgenic rice plants was significantly higher than that in ZH11 (Figure 7B) after drought treatment. In addition, the water loss rate of transgenic rice leaves after the drought treatment was significantly lower (Student’s *t*-test, *p* < 0.05) than that of ZH11, which further verified the above phenotype (Figure 7C). Finally, we concluded that *ZmHsf05* can indeed significantly improve the drought tolerance of transgenic rice.

### 3.7. Overexpression of ZmHsf05 Increases Sensitivity to Exogenous ABA Treatment

ABA is an essential hormone involved in plant responses to drought stress [12]. To test whether *ZmHsf05* regulated drought stress in an ABA-dependent manner, germination and post-germination tests were implemented. Seeds of ZH11 and transgenic rice were germinated in H_2_O (control) and 2 µM of ABA solution, respectively. Under control conditions, transgenic lines showed similar germination times and germination rates to ZH11 (Figure 8A,B), whereas, after treatment with 2 µM of ABA, the germination time was significantly delayed and the germination rate was significantly decreased in transgenic rice lines, compared with ZH11 (Figure 8C,D). Then, under normal circumstances, we selected rice seeds with consistent germination rates and estimated their sensitivity to different concentrations of ABA. After treatment for 6 days, we found that the shoot growth of rice with *ZmHsf05* overexpression was significantly decreased compared to that of wild-type ZH11 under 5 µM of ABA and 10 µM of ABA treatment (Figure 8E–F). The results indicate that the application of exogenous ABA strongly inhibited the shoot growth in the rice with transgenic *ZmHsf05* overexpression compared with the ZH11.

These data indicated that the overexpression of *ZmHsf05* in rice could increase the sensitivity of transgenic rice to exogenous ABA during seed germination and shoot growth.

### 3.8. Expression Analysis of Stress-Related Genes in ZmHsf05 Transgenic Rice

In order to further study the molecular mechanism of drought tolerance provided by overexpressing the *ZmHsf05* in transgenic rice, several genes responsive to stress were selected for the analysis of their transcription levels. As shown in Figure 9, a key gene in ABA synthesis (*OsNCED3, Os07g0154100*) [16], an abscisic stress-ripening protein5 gene which may be involved in the regulation of plant growth and drought tolerance (*OsASR5, Os11g0167800*) [13]; a late embryogenesis abundant (LEA) gene (*OsLEA3, Os06g0324400*) [50], and an anther-specific aspartic protease involving in tapetal programmed cell death (*OsAP37, Os04g0448500*) [51] were selected. Their expression levels were investigated in ZH11 and transgenic lines by RT-qPCR before and after the drought stress was initiated. We found that after drought treatment, the expression levels of *OsNCED3*, *OsASR5*, *OSLEA3*, and *OsAP37* were significantly elevated in transgenic rice compared with those in wild-type rice ZH11 (Figure 9).

## 4. Discussion

*Hsfs* are common among different eukaryotic organisms and play central roles in the plant response network [17,23,28]. Compared with those in other eukaryotes, *Hsf* families in plant species, especially in angiosperm, exhibited an expanded and varied number of genes, indicating that *Hsfs* may be involved in a more sophisticated and highly regulated transcriptional system allowing plants to cope with adverse environmental stresses [28,36]. *Hsfs* are made up of constitute three subfamilies-namely, Class A, Class B, and Class C [27]. Class A *Hsfs* were reported in transcriptional activation to various stress responses, whereas, Class B Hsfs served as repressors of gene expression or transcriptional coactivators with Class A genes. It has been reported that *Hsfs* genes are maintained in inactive states, forming complexes with heat shock proteins [52]. Upon stress, *Hsf* genes can be released from the complex and activated, eventually binding to the promoters of Hsf-responsive genes via conserved binding motifs to transcriptionally regulate types of stress-responsive genes [23,27,29,30,31]. In the present study, ZmHsf05 was found to belong to the Class A2 family, showing a clear homology to OsHsfA2e (Figure 1B). ZmHsf05 was found to harbor conserved DBD, OD, NLS and AHA domains. Consistent with the existence of the NLS and AHA domains, ZmHsf05 has been proven to be located in the nucleus in maize protoplasts and to exhibit auto activation activity in yeast. Moreover, ZmHsf05 was upregulated when the seedlings were exposed to diverse types of stresses. All these results indicated that ZmHsf05 could be a promising transcriptional activator for enhancing adaptability to diverse types of stress tolerance.

In recent years, some maize heat shock transcription factors have been reported to respond to various abiotic stresses such as *ZmHsf01* [38] and *ZmHsf04* [37]. *OsHsfA2e*, a homologous gene of *ZmHsf05*, has been reported to be able to respond to various environmental stresses [53]. In the present study, *ZmHsf05* was induced by various environmental stresses, including high temperature, drought (desiccation), high salt levels and ABA (Figure 2), suggesting the versatile functions of ZmHsf05, similar to with its orthologous genes *OsHsfA2e*. To decipher its functional characterization, *ZmHsf05* was overexpressed in rice. In the present study, transgenic rice lines overexpressing *ZmHsf05* clearly exhibited a better phenotypic performance under conditions of severe drought stress (Figure 6 and Figure 7A). Furthermore, *ZmHsf05* transgenic lines accumulated more proline, and showed lower leaf water loss rate compared to wild type plants in drought condition. With less water loss, plants could be relatively resistant to drought stress. Additionally, proline is an important kind of compatible osmolytes and could compromise cellular impairment upon osmotic or drought stress [46]. Thus, we can conclude that *ZmHsf05* improves the ability to reduce water loss from leaves and stimulates the proline accumulation to maintain cellular osmotic balance upon drought or osmotic stress. Thus, *ZmHsf05* was found to be a positive regulator in response to drought stress and enhance drought tolerance (Figure 6 and Figure 7A).

ABA acts as a central regulator in drought stress, seed germination, and relative water loss. Thus, we further investigated whether *ZmHsf05* functions through ABA-dependent pathways or not. As expected, rice overexpressing *ZmHsf05* was found to be more sensitive to ABA than ZH11, with delayed seed germination courses and a shorter shoot height under ABA treatment (Figure 8). *OsNCED3* (Os07g0154100), encoding 9-cis-epoxycarotenoid dioxygenase, which was considered to be the key enzyme in the ABA synthesis pathway [16,54], was significantly upregulated in *ZmHsf05* transgenic rice (Figure 9). Moreover, the mRNA expression level of OsLEA3 (Os06g0324400) [51], a key factor in ABA-induced antioxidant defense, was significantly higher than that in *ZmHsf05* transgenic rice compared to ZH11. We also found that OsASR5, which plays multiple roles in response to drought stress by regulating the ABA biosynthesis pathway, was greatly changed upon drought stress. Collectively, we can state that ZmHsf05 promotes plant sensitivity to ABA and positively regulates the transcription level of ABA- and stress- responsive genes, strongly indicating *ZmHsf05* positively regulates drought stress response in an ABA-dependent manner.

As most important crops in the world, the stabilization of rice and maize yield would be of great benefit for food security. *ZmHsf05* could be a promising drought-resistant gene for rice and maize breeding. Ectopic overexpression of *ZmHsf05* gene in rice may help us to acquire a modest comprehensive knowledge about the molecular function of *ZmHsf05* gene in maize. It is hoped that our study could provide a theoretical basis for the further functional characterization of *Hsf* genes in maize.

## 5. Conclusions

In general, the present study aims to investigate the molecular characterization and functions of *ZmHsf05*, a member of the A2 *hsf* genes in maize. Results showed that *ZmHsf05* located in the nucleus and has transactivation ability in yeast. Moreover, the mRNA level of *ZmHsf05* was affected by diverse abiotic stress. Overexpressing *ZmHsf05* in rice could significantly promote the drought tolerance of transgenic rice and significantly increased sensitivity to ABA treatment. Additionally, the transcription levels of ABA- and drought- responsive genes were significantly increased after drought stress in transgenic rice. Overall, *ZmHsf05* was suggested to positively regulate plant drought tolerance in an ABA-dependent manner. The present study would provide demonstrates a promising gene resource for rice and maize breeding. Lastly, it is hoped that this study can provide a theoretical basis for the further functional characterization of *Hsf* genes in maize and contribute to the improvement of global crop yield security.

## Figures and Tables

**Figure 1 genes-12-01568-f001:**
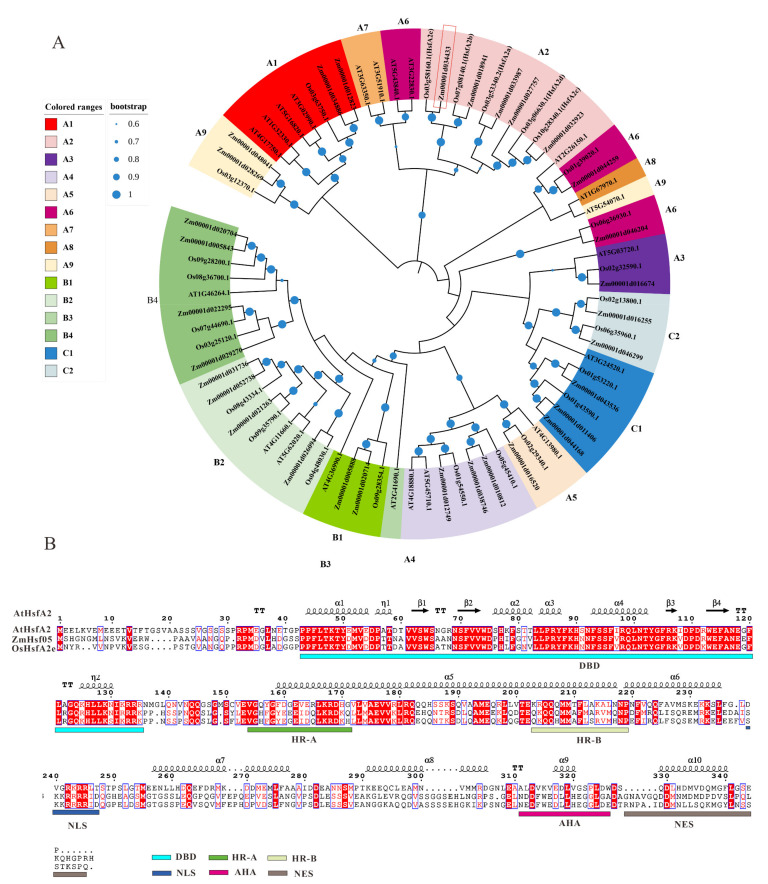
Homologous analysis of heat shock transcription factor 05 (ZmHsf05) with Hsfs in maize (*Zea mays*), rice (*Oryza sativa*) and *Arabidopsis thaliana*. (**A**) The phylogenetic tree was constructed using the neighbor-joining method. Heat shock transcription factors (Hsfs) were classified into different sub-classes according to their annotation in the Uniprot website and marked with a different color. Nodes with bootstrap values larger than 0.6 are shown by blue circles. ZmHsf05 is emphasized by a red box. Zm, Os, and AT is short for maize, rice and Arabidopsis thaliana, respectively. (**B**) Multiple sequences alignments of ZmHsf05, AtHsfA2 and OsHsfA2e. Identical and similar residues are boxed in red and white, respectively. Domain composition and position were marked with colored boxes below the alignments (DBD, DNA binding domain; OD, oligomerization domain, NLS, nuclear localization signal; AHA, aromatics hydrophobic and acidic amino acid residues; NES, nuclear export signal). The predicted second structure of AtHsfA2 was present above the alignments (helix, squiggles, β-strands, arrows; turns, TT letters).

**Figure 2 genes-12-01568-f002:**
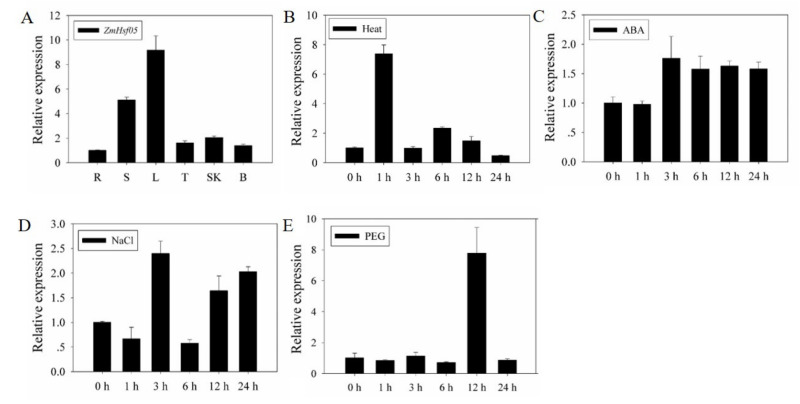
Expression patterns analysis of *ZmHsf05* in maize. (**A**) Tissue-specific expression of *ZmHsf05* gene in maize under normal conditions. R (root); S (stem); L (leaf); T (tassel); SK (corn silk); B (bract). (**B**) Expression patterns of *ZmHsf05* in leaves under 42 °C. (**C**) Expression patterns of *ZmHsf05* in leaves under 100 μM ABA. (**D**) Expression patterns of *ZmHsf05* in leaves under 200 mM NaCl. (**E**) Expression patterns of *ZmHsf05* in leaves under 20% PEG6000. The *ZmActin-1* gene was used as the internal control for normalization. Vertical bars indicate means ± SEs (*n* = 3).

**Figure 3 genes-12-01568-f003:**
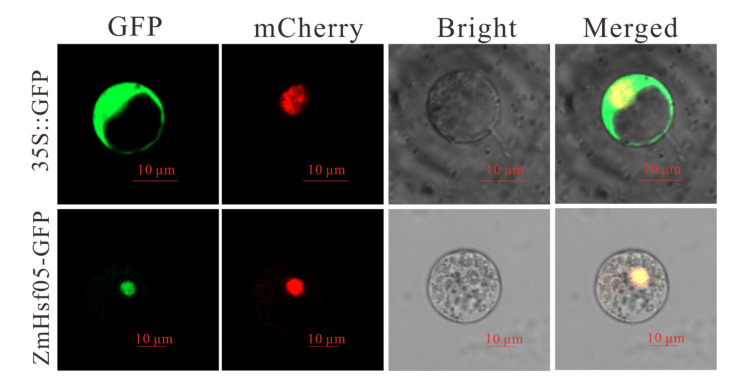
Subcellular localization of the ZmHsf05 protein. Subcellular localization of the ZmHsf05 protein in maize protoplast cells. The recombinant fluorescent proteins ZmHsf05 and GFP are localized in the nucleus, overlapped with the red fluorescent protein expressed from nuclear localization signaling protein (NLS). The 35S:GFP is used as a blank control, and ZmHsf05-GFP is the subcellular localization fusion vector of ZmHsf05. The green is the GFP protein signal, and the red fluorescence represents the nuclear localization signaling protein (NLS). (35S, CaMV 35S promoter; GFP, green fluorescent protein).

**Figure 4 genes-12-01568-f004:**
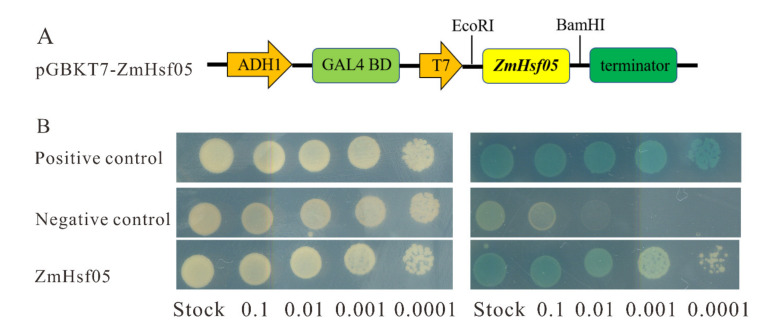
Transactivation activity analysis of *ZmHsf05*. (**A**) Schematic diagram of the fused vector used in the experiment. The full length of ZmHsf05 cDNA to the GAL4 DNA-binding domian (GAL4 BD) coding region in the fused vector. Expression of ZmHsf05 was driven by T7, while expression of GAL4 BD was driven by promoter of ADH1 (Alcohol dehydrogenase 1) (**B**) The result of the transactivation activity analysis. The yeast was transformed with positive control, negative control and pGBKT7-ZmHsf05 fused vectors, respectively, and grown on SD/-Trp/-His/-Leu/-X-α-gal triple deficiency medium. Positive control: pGBKT7-53+pGADT7-T, Negative control: pGBKT7.

**Figure 5 genes-12-01568-f005:**
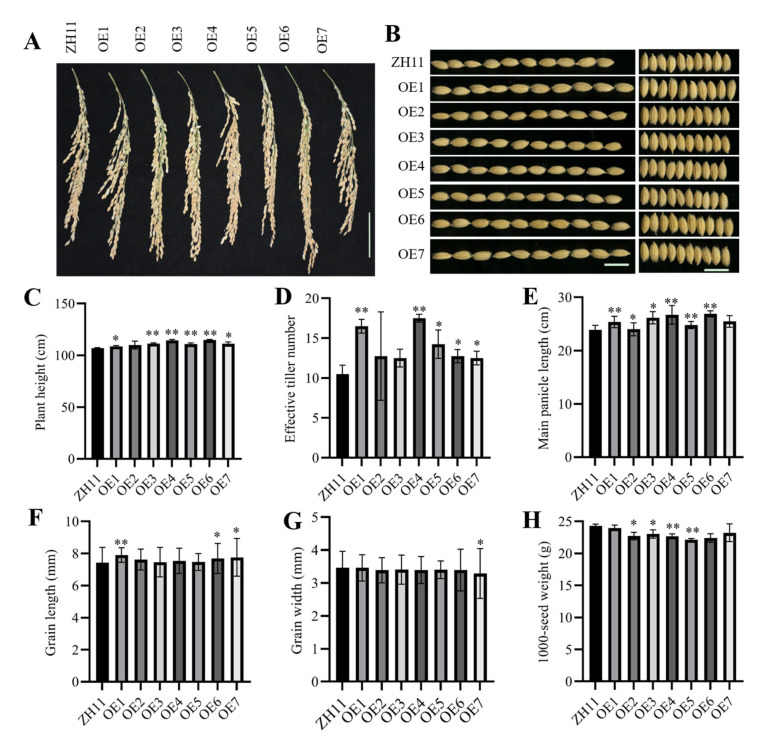
Overexpression of *ZmHsf05* in rice alters agronomical traits in field trials. Wild-type rice variety ZH11 (Zhonghua 11) and seven independent transgenic rice lines were used (OE1, OE2, OE3, OE4, OE5, OE6 and OE7). (**A**) Mature ZH11 and transgenic rice panicles. (Scale bar, 5 cm.) (**B**) Seed length (left panel), and seed width (right panel) for ten seeds in a row (Scale bar, 1 cm.). (**C**) Plant height (*n* = 5). (**D**) Effective tiller number (*n* = 5). (**E**) Main panicle length (*n* = 5). (**F**) Grain length (*n* = 90). (**G**) Grain width (*n* = 95). (**H**) 1000-seed weight (*n* = 3) were evaluated for overexpression transgenic lines and wild-type plants, respectively. Students’ *t*-test was performed for statistics analysis, * denotes *p* < 0.05, ** denotes for *p* < 0.01, n represents the number of repetitions that have been determined.

**Figure 6 genes-12-01568-f006:**
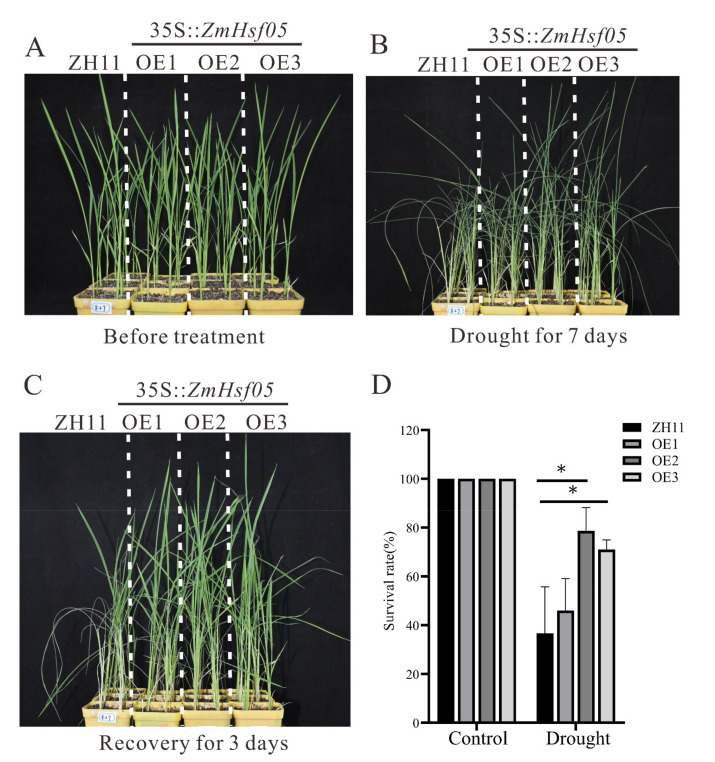
Drought tolerance analysis of rice with *ZmHsf05* overexpression in soil. Three transgenic rice *ZmHsf05* overexpression lines (OE1, OE2, and OE3) and wild-type ZH11 were cultured in soil. Rice were grown for three weeks at normal conditions (**A**), the watering was stopped and the drought treatment was carried out for 7 days (**B**). After the appearance of the phenotype, we re-cultivated the rice seedlings by watering them for 3 days (**C**). (**D**) Statistics of the survival rate of overexpression lines and WT after drought treatment. Vertical bars indicate means ± SEs (*n* = 3). *, *p* < 0.05 (Student’s *t*-test).

**Figure 7 genes-12-01568-f007:**
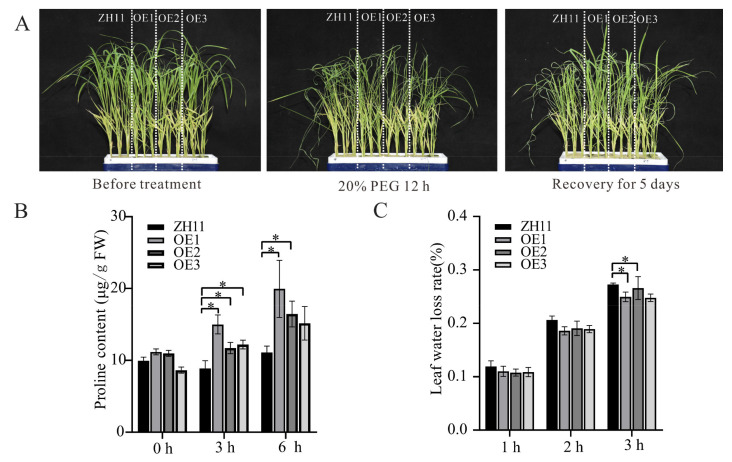
Simulated drought tolerance analysis of rice with *ZmHsf05*-overexpression rice. (**A**) Phenotypic performance of transgenic rice before and after simulated drought treatment. Three transgenic rice *ZmHsf05*-overexpression lines (OE1, OE2 and OE3) and wild-type ZH11 lines were cultured in 5-times-diluted Hoagland nutrient solution supplemented with 20% PEG6000. (**B**) Three-week-old rice seedlings were subjected to 25% PEG6000 simulated drought treatment for 3 h and 6 h to determine the content of proline. Three biological replicates were used for each experiment. Vertical bars indicate means ± SEs (*n* = 3). *, *p* < 0.05 (Student’s *t*-test). (**C**) Three-week-old rice seedling leaves were subjected to 1 h, 2 h and 3 h in vitro leaf water loss rate measurements.

**Figure 8 genes-12-01568-f008:**
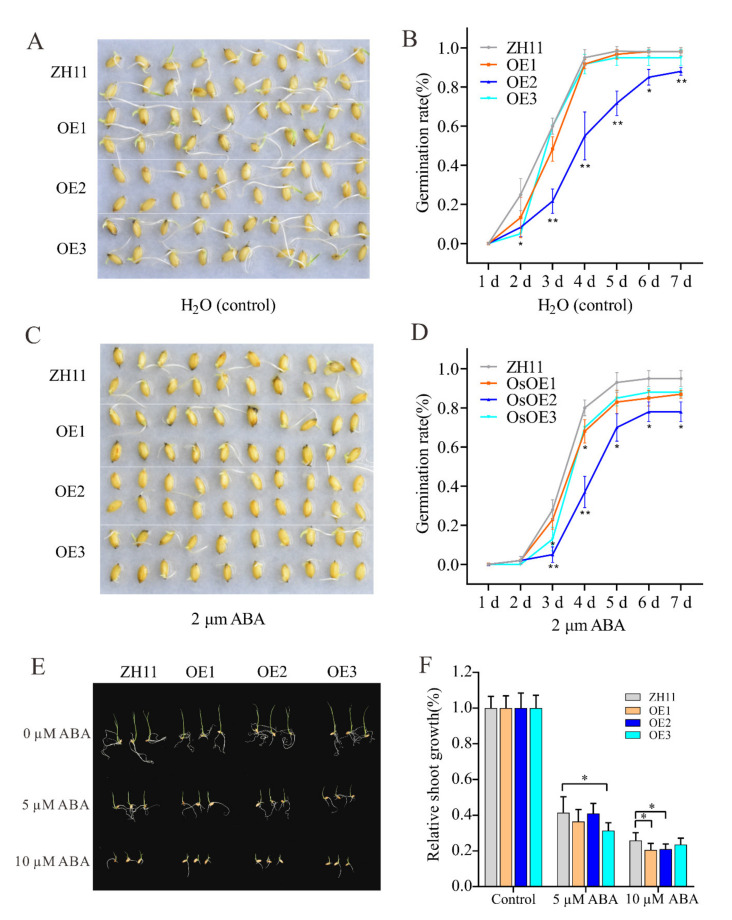
*ZmHsf05* influences seed germination and shoot growth. Each experiment utilizes three transgenic rice lines (OE1, OE2, and OE3) and wild-type ZH11. (**A**) Germination performance of transgenic rice and ZH11 in H_2_O at seven days after initiation. (**B**) Calculation and statistical analysis of the germination rate of transgenic and ZH11 rice seed in water. A seed was suggested to germinate when the length of radicle was longer than half of the seed. (**C**) Germination performance of transgenic rice and ZH11 in water containing 2 µM of ABA. (**D**) Germination rate of transgenic rice and ZH11 rice seed in water containing 2 µM of ABA. (**E**) The shoot growth of transgenic rice showed hypersensitivity to exogenous ABA. (**F**) Relative shoot length of transgenic rice and ZH11 with the addition of exogenous ABA. Three biological replicates were used for performed for each experiment. Representative graphs are shown (*n* = 20 seeds in each experiment). *, *p* < 0.05 and **, *p* < 0.01 (Student’s *t*-test).

**Figure 9 genes-12-01568-f009:**
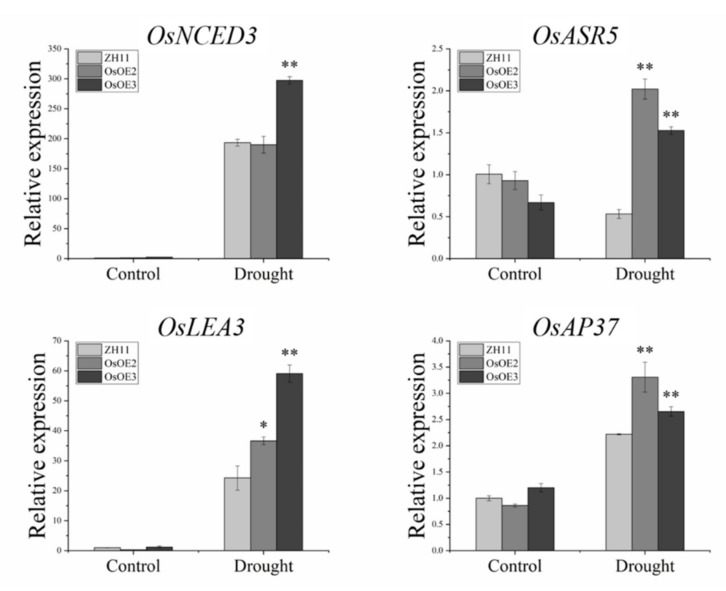
Expression analysis of ABA-related and drought-related genes in transgenic rice after drought treatment. Three biological replicates were used for each experiment. The *OsActin-1* gene was used as the internal control for normalization. Vertical bars indicate means ± SEs (*n* = 3). *, *p* < 0.05; **, *p* < 0.01 (Student’s *t*-test).

## Data Availability

We obtained the full-length CDS sequence of the *ZmHsf05* gene from the online database Phytozome (https://phytozome.jgi.doe.gov/pz/portal.html accessed on 18 April 2014). We use the online website ExPASy (https://web.expasy.org/compute_pi/ accessed on 1 July 2020) to predict translation of the protein of the *ZmHsf05* gene. All other data supporting the results are included within the article and its Additional files.

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
