# Peer review of "Ectopic Overexpression of Maize Heat Stress Transcription Factor ZmHsf05 Confers Drought Tolerance in Transgenic Rice"

_genes, 2021, doi:10.3390/genes12101568_

Round 1

Reviewer 1 Report

In their manuscript, Si et al. analyse the impact of the maize heat shock factor ZmHsf05 on drought tolerance in maize and in heterologous system (overexpression in rice plants). Initially described as regulators of the transcription in response to the heat, the HSFs have found to be involved in many stress responses including drought. In this respect, the investigations of HSFs as regulators of diverse stress responses is of importance. The work under review however represents rather short report than a full study. Some results are expectable (nuclear localization of the protein). The most intriguing results are likely increased yield in the transgenic rice under non-stressed conditions. But the reason for this increase is not clear. Grain yield is the cumulative function of grain number (but not spike number or spike length!) and grain weight. Neither of these parameters were measured. Moreover, the total grain weight could be not affected or even decreased because of the indications that the transgenic grains were longer but thinner. Therefore, the mechanism of increased yield is not clear either under normal or stressed conditions. The possible explanations of the reasons how the overexpression of ZmHsf05 can lead to higher yield under normal growth condition would be of interest.

My more specific comments are below.

Major concerns

The Title is not appropriate. The authors did not show that ZmHSF05 is a key player in drought tolerance but simply that the TF is responsive to drought and may regulate plant’s response to drought stress. Similarly, the authors have solely shown the TF reacts on ABA treatment but did not show how. No one gene of ABA-dependent pathway was analysed.

Introduction. In an ideal case, the introduction should summarize the existing knowledge about the topic and highlight the gaps, which the authors intend to fill with their investigation. In this respect, the introduction is failed. The HSF family is well described in plants; however, many important publications were not referred (e.g., Scharf et al. 2012; Li, Zhang et al. 2015; Manna et al. 2021, and others). It is helpful to mention that HSFs are encoded by many genes (how many in maize?) and are classified into several categories (Kotak et al., 2004). I would recommend to rewrite the introduction by focussing more on current knowledge about HSFs and their role in drought (and other stresses) instead giving a bit information from everything about drought.

English must be significantly improved.

What do the authors wanted to analyse by transactivation assay?

Figures

To my opinion, the Figure 1A is superfluous because the sequences of both cDNA and protein are easily searchable in open databases by gene ID. The domain structure is typical for this kind of TFs and is described in the Results. Figure 1b would be more informative when the phylogenetic tree of the whole maize HSF family would be compared with that of e.g. rice and Arabidopsis instead to present the homology alignment with randomly selected HSF proteins. Phylogenetic tree allows to determinate the evolutionary position of the MzHsf05.

Figure 2 B – E: What tissue was analysed in these experiments? Please also exactly determinate the number of repetitions (n).

Figure 3B: the photos of protoplasts are too small and are not allow to see intracellular structures. Please remake. The scales are missing.

Figure 5. Please define n here, but also in all other cases. The indication n > 3 is not satisfactory.

Figures 6A, 7A, 8A, C and E. The phenotypes are barely visible. Please, remake the figures to show better the response of different lines (especially controls).

Figure 6B. What do the white bars show under control? Did every transgenic line have its own control?

The discussion is very weak. The authors should bring their observation in the context of the existing knowledge und discuss unexpected or intriguing points.

Supplemental Fig. 2. The bars in B and D imply longer or thinner grains, respectively; however, the pictures of grains (A and B) do not support this.

Other remarks:

All abbreviations (e.g., DREB, WRKY, bZIP on the line 49, but also everywhere in the text) should be written by full names when first mentioned.

lines 35-36: the sentence starting with “Though various yield…” has no sense and must be rewritten.

The subchapter 2.2 from Material and methods is fully confusing and must be completely rewritten.

The subchapter 2.3 is also incomplete and not clear. Were plants treated with PEG for 4 weeks or 3-6 hours?

The reference gene for qRT-PCR is not mentioned in MandM

The subchapter 2.6 is not clearly described.

Line 231: what does it mean “this gene has no output signal domain”?

line 251. What was the reason to analyse the expression by semi-quantitative method while the more improved method (qRT-PCR) was also applied?

Author Response

Comments and Suggestions for Authors

In their manuscript, Si et al. analyze the impact of the maize heat shock factor ZmHsf05 on drought tolerance in maize and in heterologous system (overexpression in rice plants). Initially described as regulators of the transcription in response to the heat, the HSFs have found to be involved in many stress responses including drought. In this respect, the investigations of HSFs as regulators of diverse stress responses is of importance. The work under review however represents rather short report than a full study. Some results are expectable (nuclear localization of the protein). The most intriguing results are likely increased yield in the transgenic rice under non-stressed conditions. But the reason for this increase is not clear. Grain yield is the cumulative function of grain number (but not spike number or spike length!) and grain weight. Neither of these parameters were measured. Moreover, the total grain weight could be not affected or even decreased because of the indications that the transgenic grains were longer but thinner. Therefore, the mechanism of increased yield is not clear either under normal or stressed conditions. The possible explanations of the reasons how the overexpression of ZmHsf05 can lead to higher yield under normal growth condition would be of interest.

Responses: Many thanks for your kind encouragement and insightful suggestions. Hsfs gene family, existing across eukaryote kingdoms, exhibited an expanded and varied number of genes in angiosperm species, indicating that Hsfs in plants may have versatile roles in a more sophisticated and highly regulated transcriptional system to cope with adverse environmental stresses. As you mentioned, Hsfs were initially recognized as regulators in response to heat, however, some Hsfs genes have been characterized for their crucial roles in response to other stresses, including drought, salinity etc. The present study attempted to characterize the molecular function of ZmHsf05 gene in maize. However, as a result of difficulty in transformation of maize, we decided to overexpress ZmHsf05 gene in rice variety ZH11 (Zhonghua 11). Ectopic overexpression of ZmHsf05 gene in rice may help us to acquire a modest comprehensive knowledge about the molecular function of ZmHsf05 gene in maize. Expectedly, ectopic overexpression of ZmHsf05 gene in rice confers increased drought tolerance compared to wild-type plants.

According to your suggestions, thousand seed weights of transgenic rice and ZH11 were also assessed in the revised manuscript. Field trials demonstrated that overexpression of ZmHsf05 in rice can promote effective tiller emergence, improve plant height, increase panicle length, modify seed morphology and slightly reduce seed weight. These results indicated that overexpressing ZmHsf05 gene in rice could alter agronomical traits in field trials. As transgenic lines have remarkably larger number of effective tillers, which suggested overexpressing ZmHsf05 in rice may have great potential to increase yield production. Similar results were observed in TaHsf2d transgenic lines. Chauhan et al. reported that overexpression of TaHsf2d gene in A. Arabidopsis could considerably enhance tolerance to heat stress (Chauhan et al., 2013). Meanwhile, TaHsf2d transgenic lines showed higher yield and biomass accumulation. However, the molecular mechanism of Hsfs in regulation of yield production largely remains elusive currently. According to your suggestions, we will continue to explore the transcriptional network that ZmHsf05 participated in regulation of drought tolerance as well as agronomic traits.

Chauhan, H., Khurana, N., Agarwal, P., Khurana, J.P., and Khurana, P. (2013). A seed preferential heat shock transcription factor from wheat provides abiotic stress tolerance and yield enhancement in transgenic Arabidopsis under heat stress environment. PloS one 8, e79577.

My more specific comments are below.

Major concerns

The Title is not appropriate. The authors did not show that ZmHSF05 is a key player in drought tolerance but simply that the TF is responsive to drought and may regulate plant’s response to drought stress. Similarly, the authors have solely shown the TF reacts on ABA treatment but did not show how. No one gene of ABA-dependent pathway was analyzed.

Responses: Thanks for your insightful comments. In the present study, transgenic rice lines overexpressing ZmHsf05 gene exhibited better phenotypic performance under severe drought stress. Moreover, we found that overexpression transgenic lines showed hypersensitive to exogenous ABA treatment. Additionally, mRNA expression level of several key genes, including OsNCED3, OsLEA3, and OsASR5, which were proposed to be involved in ABA biosynthesis, and signaling transduction pathway, were significantly higher than that in ZH11 upon drought stress. Therefore, we speculated that ZmHsf05 could improve the drought resistance in an ABA-dependent manner. According to your suggestions, the title has been changed to “Ectopic overexpression of maize heat stress transcription factor ZmHsf05 confers drought tolerance in transgenic rice”.

Introduction. In an ideal case, the introduction should summarize the existing knowledge about the topic and highlight the gaps, which the authors intend to fill with their investigation. In this respect, the introduction is failed. The HSF family is well described in plants; however, many important publications were not referred (e.g., Scharf et al. 2012; Li, Zhang et al. 2015; Manna et al. 2021, and others). It is helpful to mention that HSFs are encoded by many genes (how many in maize?) and are classified into several categories (Kotak et al., 2004). I would recommend to rewrite the introduction by focussing more on current knowledge about HSFs and their role in drought (and other stresses) instead giving a bit information from everything about drought.

Responses: Many thanks for your valuable suggestion. The introduction part has been rewritten thoroughly. Sufficient information about Hsfs family was provided, and its crucial roles in helping plants to survive from adverse environmental stresses were emphasized in the revised manuscript. Additionally, more important publications were cited.

English must be significantly improved.

Responses: Many thanks for your valuable suggestion. The English language has been substantially improved by a professional English editing service.

What do the authors want to analyze by transactivation assay?

Responses: Many thanks for your comments. Class A Hsfs are promising transcriptional activators with conserved AHA domain. ZmHsf05 gene belongs to Class A2 type Hsf subfamily. ZmHsf05 protein harbors typically conserved DBD, OD, NLS and AHA domain. Thus, the activator function of ZmHsf05 is essential to analysis. Plants and yeast showed conservation of activator function and the interacting components of the transcriptional machinery. In some papers, transactivation assay with Gal4DBD fusion proteins has been applied to test the activator potential of transcription factors in plants (Yang et al., 2019; Yokotani et al., 2008; Zhou et al., 2019). Thus, a transactivation assay was also conducted in the present study. Results showed ZmHsf05 can be transactivated in yeast, suggesting it has a potential to bind to DNA elements to activate downstream stress-responsive genes.

Yang, S., Xu, K., Chen, S., Li, T., Xia, H., Chen, L., Liu, H., and Luo, L. (2019). A stress-responsive bZIP transcription factor OsbZIP62 improves drought and oxidative tolerance in rice. BMC Plant Biol 19, 260.

Yokotani, N., Ichikawa, T., Kondou, Y., Matsui, M., Hirochika, H., Iwabuchi, M., and Oda, K. (2008). Expression of rice heat stress transcription factor OsHsfA2e enhances tolerance to environmental stresses in transgenic Arabidopsis. Planta 227, 957-967.

Zhou, Y., Zhu, H., He, S., Zhai, H., Zhao, N., Xing, S., Wei, Z., and Liu, Q. (2019). A Novel Sweetpotato Transcription Factor Gene IbMYB116 Enhances Drought Tolerance in Transgenic Arabidopsis. Frontiers in plant science 10, 1025.

To my opinion, the Figure 1A is superfluous because the sequences of both cDNA and protein are easily searchable in open databases by gene ID. The domain structure is typical for this kind of TFs and is described in the Results. Figure 1b would be more informative when the phylogenetic tree of the whole maize HSF family would be compared with that of e.g. rice and Arabidopsis instead to present the homology alignment with randomly selected HSF proteins. Phylogenetic tree allows to determinate the evolutionary position of the MzHsf05.

Responses: Many thanks for your insightful suggestions. According to your suggestions, a neighbor-joining tree was constructed with all Hsfs proteins from maize (Zea mays), rice (Oryza sativa) and Arabidopisis thaliana. In the phylogenetic tree, ZmHsf05 is orthologous with OsHsfA2e and AtHsfA2. Subsequently, protein sequences of ZmHsf05, OsHsfA2e, and AtHsfA2 were aligned. The domain composition and secondary structure of ZmHsf05 was further shown according to previous reports of OsHsfA2e and AtHsfA2.

Figure 2 B – E: What tissue was analyzed in these experiments? Please also exactly determinate the number of repetitions (n).

Responses: Thanks for your comment. Leaves of three-leave stage seedlings and maize inbred line B73 were sampled in the experiments of induced expression pattern analysis. Each tissue was repeatedly sampled from three individuals as three biological duplicates. These mistakes were modified in both the figure legend and methods

Figure 3B: the photos of protoplasts are too small and are not allow to see intracellular structures. Please remake. The scales are missing.

Responses: Many thanks for your valuable suggestions. Figure 3 has been remake in the revised manuscript.

Figure 5. Please define n here, but also in all other cases. The indication n > 3 is not satisfactory.

Responses: Many thanks for your valuable suggestions. The number of repetitions has been exactly determinate in the revised manuscript.

Figures 6A, 7A, 8A, C and E. The phenotypes are barely visible. Please, remake the figures to show better the response of different lines (especially controls).

Responses: Many thanks for your valuable suggestions. In order to better display the phenotypes, all figures have been remake in the revised manuscript.

Figure 6B. What do the white bars show under control? Did every transgenic line have its own control?

Responses: Many thanks for your valuable suggestions. We apologize for this mistake. In the present study, one control was used for survival rate analysis. We have remake Figure 6B in the revised manuscript.

The discussion is very weak. The authors should bring their observation in the context of the existing knowledge und discuss unexpected or intriguing points.

Responses: Many thanks for your valuable suggestions. The discussion part has been rewritten thoroughly in the revised manuscript.

Supplemental Fig. 2. The bars in B and D imply longer or thinner grains, respectively; however, the pictures of grains (A and B) do not support this.

Responses: Thank you for your valuable suggestions. The pictures of grains were remake and moved to Figure 5.

Other remarks:

All abbreviations (e.g., DREB, WRKY, bZIP on the line 49, but also everywhere in the text) should be written by full names when first mentioned.

Responses: Many thanks for your valuable suggestions. Improper usage of abbreviations has been corrected thoroughly in the revised manuscript.

lines 35-36: the sentence starting with “Though various yield…” has no sense and must be rewritten.

Responses: Thanks for your suggestion. This sentence has been rewritten in the revised manuscript.

The subchapter 2.2 from Material and methods is fully confusing and must be completely rewritten.

Responses: Thanks for your suggestion. In the revised manuscript, the subchapter 2.2 and 2.2 has been merged and completely rewritten.

The subchapter 2.3 is also incomplete and not clear. Were plants treated with PEG for 4 weeks or 3-6 hours?

Responses: Thanks for your suggestion. In the revised manuscript, the subchapter 2.2 and 2.3 has been merged and completely rewritten. The sentence you mentioned has been rewritten as follows:

“For the analysis of stress-related gene expression, rice seedlings were grown in Hoagland nutrient solution for three weeks and then treated with 20% PEG. After treatment for 3 hours and 6 hours, leaves of ZH11 and transgenic lines were rapidly sampled.”

The reference gene for qRT-PCR is not mentioned in MandM

Responses: Thanks for your suggestion. For standardized analysis, ZmActin1 (Zm00001d010159) and OsActin1 (Os03g50885) were used as controls, respectively. We have modified this in the revised manuscript.

The subchapter 2.6 is not clearly described.

Responses: Thanks for your suggestion. This subchapter 2.6 has been clarified in the revised manuscript.

Line 231: what does it mean “this gene has no output signal domain”?

Responses: According to the former protein sequence analysis of ZmHsf05, we found that ZmHsf05 did not harbor a nuclear export signal domain (NES) similar with those in tomato or Arabidopsis. Thus, we inferred that this gene has no output signal domain. Whereas, in the revised manuscript, domain composition of ZmHsf05 was carefully investigated by the multiple alignments of ZmHsf05 with OsHsfA2e and AtHsfA2. A candidate NES domain was identified. Thus this sentence and related sentences had been deleted or amended.

line 251. What was the reason to analyze the expression by semi-quantitative method while the more improved method (qRT-PCR) was also applied?

Responses: ZmHsf05 was ectopic overexpressed in rice and the expression level of ZmHsf05 could be detected in the wild-type plant ZH11, which was generally used as a control in RT-qPCR. Thus, we believed that semi-quantitative method was more accurate to confirm the expression of ZmHsf05 in transgenic rice. The RT-qPCR experiments in this study applied expression level of ZmHsf05 in OE1 as a control, which could only simply assess the expression level variations among different transgenic lines.

Reviewer 2 Report

There are extensive English language errors which made the manuscript difficult to follow.  I started to mark these but there are too many throughout the manuscript for me as a reviewer to correct.  Be consistent with capitalizations in the title and elsewhere (e.g., abscisic acid should be lower case) and always italicize gene names. 

The science, as I am able to follow it, seems sound and the results are interesting.  It's not completely clear as written whether your target is improving drought tolerance of maize or rice.  Avoid using terms like "water retaining capacity" unless you define it precisely and measure it specifically.  What exactly do you mean by "drought tolerance" and how is this different (or is it?) from "drought resistance"?

More details should be provided in figure legends such that all terms are defined (e.g., functional domains like "DBD" in 1st part of Fig1 and species names in the 2nd part of Fig.1; what are the time points in Fig2B-E; construct parts in Fig3A, which is "control vector"; parts of the vector in Fig4A; expand explanation of Fig4B; list the transgenic rice lines in parentheses in Figs 5, 6 and 8).

Overall, it is potentially a good contribution and scientifically sound, but unfortunately, it is not clearly presented, though all the supportive evidence seems to be present and organized fairly well.  The English errors muddy your message to the point where it is unclear.  

Author Response

Comments and Suggestions for Authors

There are extensive English language errors which made the manuscript difficult to follow.  I started to mark these but there are too many throughout the manuscript for me as a reviewer to correct.  Be consistent with capitalizations in the title and elsewhere (e.g., abscisic acid should be lower case) and always italicize gene names.

Responses: Many thanks for your kindness and valuable comments. We are so sorry for these careless errors and improper usage of abbreviations. We have tried every effort to make sentences clear. Moreover, the English language has been substantially improved by a professional English editing service in the revised manuscript. We hope it would be easier to follow.

The science, as I am able to follow it, seems sound and the results are interesting.  It's not completely clear as written whether your target is improving drought tolerance of maize or rice.  Avoid using terms like "water retaining capacity" unless you define it precisely and measure it specifically.  What exactly do you mean by "drought tolerance" and how is this different (or is it?) from "drought resistance"?

Responses: Many thanks for your kind encouragement. The present study attempted to characterize the molecular function of ZmHsf05 gene in maize. However, as a result of difficulty in transformation of maize, we decided to overexpress ZmHsf05 gene in rice variety ZH11. Ectopic overexpression of ZmHsf05 gene in rice may help us to acquire a modest comprehensive knowledge about the molecular function of ZmHsf05 gene in maize. To make this issue more clearly, the title has been changed to “Ectopic overexpression of maize heat stress transcription factor ZmHsf05 confers drought tolerance in transgenic rice”. Some explanations were also provided in the introduction and discussion parts.

Indeed, the item “Water retaining capacity” was improper. We have changed “water retaining capacity” into “water loss rate” in the revised manuscript. Drought resistance were proposed to be achieved by either increasing water uptake or reducing water losses. Whereas, drought tolerance maintains the physiological processes under drought stress and produces higher economic yield. Therefore, we preferred to use the item “drought tolerance” instead of “drought resistance”.

More details should be provided in figure legends such that all terms are defined (e.g., functional domains like "DBD" in 1st part of Fig1 and species names in the 2nd part of Fig.1; what are the time points in Fig2B-E; construct parts in Fig3A, which is "control vector"; parts of the vector in Fig4A; expand explanation of Fig4B; list the transgenic rice lines in parentheses in Figs 5, 6 and 8).

Responses: Many thanks for your valuable suggestions. Figures were remake and all of Figure legends were also clarified and updated with more details in the revised manuscript.

Overall, it is potentially a good contribution and scientifically sound, but unfortunately, it is not clearly presented, though all the supportive evidence seems to be present and organized fairly well.  The English errors muddy your message to the point where it is unclear.

Responses: Many thanks for your encouragement and valuable comments. We apologize for these rudeness errors. The manuscript has been extensively rewritten. Moreover, the grammar and language has been substantially improved by a professional English editing service in the revised manuscript.
